# Wavefront dislocations reveal the topology of quasi-1D photonic insulators

Clément Dutreix [1 ✉], Matthieu Bellec[2], Pierre Delplace [3] & Fabrice Mortessagne [2 ✉]

Phase singularities appear ubiquitously in wavefields, regardless of the wave equation. Such topological defects can lead to wavefront dislocations, as observed in a humongous number of classical wave experiments. Phase singularities of wave functions are also at the heart of the topological classification of the gapped phases of matter. Despite identical singular features, topological insulators and topological defects in waves remain two distinct fields. Realising 1D microwave insulators, we experimentally observe a wavefront dislocation – a 2D phase singularity – in the local density of states when the systems undergo a topological phase transition. We show theoretically that the change in the number of interference fringes at the transition reveals the topological index that characterises the band topology in the insulator.

[1] Univ. Bordeaux, CNRS, LOMA, UMR 5798, F-33400 Talence, France. [2] Université Côte d'Azur, CNRS, Institut de Physique de Nice, Nice, France. [3] Univ Lyon, Ens de Lyon, Univ Claude Bernard, CNRS, Laboratoire de Physique, Lyon, France. ✉email: clement.dutreix@u-bordeaux.fr; fabrice.mortessagne@univ-cotedazur.fr

avefront dislocations are a fundamental and ubiqui-
tous wave phenomenon that originates from an
indetermination of the phase of a wavefield, where the
amplitude of the wave vanishes. Since the seminal work of John
Nye and Michael Berry in 1974[1], it was realised that such topo-
logical defects could emerge in any wavefield irrespectively of its
physical nature or its dispersion relation. Wavefront dislocations
have been observed from the physics of fluids, sound, electro-
magnetism to oceanic tides and astronomy, and even led to the
birth of a whole research field known as singular optics[2–7]. In
quantum mechanics, wavefront dislocations have been predicted
in connection to the Aharonov–Bohm effect[8,9], and they have
been observed recently in the electron density of graphene as a
manifestation of the wave-function Berry phase[10,11]. In parallel,
topology also spread in condensed matter physics, giving rise to
the field of topological phases of matter[12,13] and its various
classical analogues such as topological photonics[14]. In this con-
text, topological properties are defined from singularities of the
wave functions delocalised in the bulk of the material. Still, apart
from the pioneering example of the quantised electric con-
ductivity in the quantum Hall phase, their experimental mani-
festations are mainly indirect, through the existence of gapless
excitations localised at the boundaries of the system. Despite this
common underlying key role of phase singularities, topological
phases and singular waves have remained two distinct fields[15].
Here we reconcile them by showing a wavefront dislocation as a
direct evidence of the phase singularity of the delocalised wave
functions and observe it through standing-wave interference in
1D microwave photonic insulators. By bridging the bulk topology
of insulators to a ubiquitous wave phenomenon, we open a
promising route to investigate quantum and classical topological
systems through real-space interference patterns.

Topological insulating systems are associated with integer-
valued numbers (topological indices) that characterise the phase
singularities of the bulk wave functions. A change of the topo-
logical index requires the spectral gap to close. Such a topological
transition is also associated with a phase singularity of the
eigenmodes. By including the parameter that controls the spectral
gap, a topological transition of a $D$ dimensional system is then
described by a singular point in a $D + 1$ dimensional parameter
space. The 1D case is particularly interesting as it allows us to
describe topological transitions with vortices appearing in 2D
parameter space. Vortices in real space are known to induce
wavefront dislocations onto an incident propagating wave[8].
Similarly, we reveal here that the vortex of the topological tran-
sition involves a quite analogue phenomenon in parameter space.
When a defect or an edge is included to a topological insulator, a
defect-induced interference pattern of bulk wave functions
emerges and abruptly changes at the singular band crossing point,
then giving rise to a wavefront dislocation in the $D + 1$ parameter
space. We find that this wavefront dislocation is accessible
experimentally through the local density of states (LDOS) and
demonstrate its existence in 1D microwave photonic insulators.
Moreover, we show that the quantised charge of the vortex, which
corresponds to the variation of the number of interference fringes
at the transition, consistently coincides with the variation in the
number of topologically-protected boundary modes inside the
spectral gap. This leads us to the experimental demonstration of
the pillar of the topological phases of matter, the bulk-boundary
correspondence.

## Results

**Realisation of the 1D photonic insulator.** In 1D, the band
topology of insulators may become nontrivial in the presence of
chiral symmetry. For lattices with translational invariance, this
concerns a class of Bloch Hamiltonians that are bipartite

$$H(k) = \begin{pmatrix} 0 & h(k) \\ h^\dagger(k) & 0 \end{pmatrix}, \qquad (1)$$

where $k$ is the dimensionless 1D quasi-momentum. An illumi-
nating illustration of Hamiltonian (1) is found in the celebrated
Su–Schrieffer–Heeger (SSH) model[16]. First introduced to describe
conducting electrons in polyacetylene, it is involved in the physics
of various chiral systems[17–22]. Here, we focus on an experimental
realisation in a microwave photonic insulator. The system con-
sists of a dimerised lattice of dielectric resonators in a microwave
cavity (see Fig. 1a, b). Each cylindrical resonator is made of
ZrSnTiO ceramics (radius $r = 3$ mm, height $h = 5$ mm, with an
index of refraction $n_r \approx 6$) and supports a fundamental
transverse-electric ($TE_1$) mode of bare frequency of 7.435 GHz.
This mode spreads out evanescently, so that the coupling strength
can be controlled by adjusting the separation distance between
the resonators[23]. The lattice consists in two coupled sublattices A
and B with staggered coupling strengths $t_1$ and $t_2$, so that $h(k) =
t_1 + t_2 e^{-ik}$ for the choice of unit cell in Fig. 1b. The corresponding
resonator separations are denoted $d_1$ and $d_2$. In our experiments,
the coupling strengths $t_{1,2}$ can be typically adjusted from 10 to

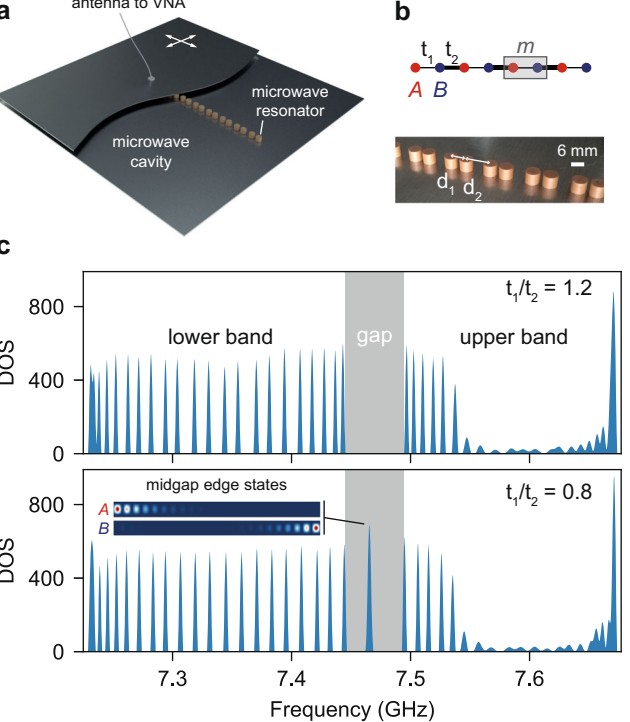

**Fig. 1 1D microwave photonic insulator. a** Lattice of dielectric resonators
inside the microwave cavity made of two metallic plates. The top plate,
which is partially shown, is movable in the plane (white arrows), so that an
antenna going through and connected to a vector network analyser (VNA)
enables the generation and resolution of the microwave signal both
spectrally and spatially (LDOS). **b** Illustration of the SSH lattice (top) and
picture of its realisation with dielectric resonators (bottom). The
corresponding resonator separations are denoted $d_1$ and $d_2$. **c** DOS of a 44-
resonator SSH lattice for $t_1/t_2 = 1.2$ ($w_> = 0$) showing two bands separated
by a frequency gap (top), and for $t_1/t_2 = 0.8$ ($w_< = -1$), revealing two
midgap modes $A$ and $B$ (bottom). The resolution of their LDOS is shown in
the inset for the resonators of sublattices A and B, respectively. The LDOS
is in arbitrary units—red (blue) colour means 1 (0). The LDOS on sublattice
A (B) is maximal on the leftmost (rightmost) site and vanishes when
moving rightward (leftward). It confirms that the midgap modes $A$ and $B$
are localised at the two ends of the lattice and fully sublattice polarised.

115 MHz which corresponds to separations $d_{1,2}$ of 16 mm and 7 mm, respectively.

The SSH model is known to display a transition between two topologically distinct insulators when varying the coupling ratio $t_1/t_2$. Its spectrum exhibits two bands given by $f_\pm(k) = \pm|h(k)|$ and whose topology relies on the quantisation of the geometrical Zak phase of the Bloch wave function in the 1D Brillouin zone (BZ)[24]. This quantisation is characterised by the winding number $w = \oint_{BZ} \nabla_k \text{Arg}[h(k)]dk/2\pi$, which leads to $w_> = 0$ and $w_< = -1$ in the two insulating regimes $t_1 \gtrless t_2$.

**Topology from localised boundary modes.** Before shedding new light on the topological transition, let us recall that in experiments, the band topology is mainly evidenced through the appearance/disappearance of midgap modes localised at the lattice boundaries, by virtue of the famous bulk-boundary correspondence[25]. Here, the bulk-boundary correspondence predicts the existence of $\mathcal{N}_A = -w_\gtrless$ ($\mathcal{N}_B = -w_\gtrless$) bound states with sublattice polarisation $A$ ($B$) at the leftmost (rightmost) edge of the crystal in Fig. 1b (see Supplementary Note 3). We report the observation of these midgap boundary modes in Fig. 1c, d. It shows the measured density of states (DOS) of the $TE_1$ waves in a lattice of 44 microwave resonators (see Supplementary Note 1). For $t_1/t_2 = 1.2$, where the bulk winding number is $w_> = 0$, the sublattice structure produces two frequency bands of 22 modes each. In contrast, when $t_1/t_2 = 0.8$, the 1D winding number switches to $w_< = -1$ and we observe two modes pinned in the gap. We then confirm that they are sublattice polarised and spatially localised at the two ends of the crystal by resolving their LDOS (inset of Fig. 1d).

The observation of midgap modes *localised* at boundaries is commonly considered as the hallmark of topological transitions, as reported in mechanical, acoustic, photonic, microwave, cold-atomic and electronic systems[26–33]. Nevertheless, the band topology is defined from the delocalised waves beyond the excitation gap. Now we present direct evidence of the topological transition through LDOS measurements of the *delocalised* waves.

**LDOS interferences of delocalised waves.** The delocalised waves correspond to resonance frequencies in the two bands $f_\pm$. Figures 2a–d represents the sublattice-resolved LDOS $\rho_{A,B}$ of the delocalised waves of the lower band $f_-$ probed in the two topological regimes. Only the leftmost half of the photonic crystal is shown, for the second half is inversion symmetric (see Supplementary Note 2). The LDOS maps consist of standing-wave interference patterns due to the lattice boundaries. We focus in particular on the number $N_{A(B)}$ of constructive-interference fringes in the LDOS of sublattice $A(B)$. For the sublattice $A$ in Fig. 2a, b, we observe that $N_A$ changes identically on each site $m$ through the topological transition. For instance, there are six constructive-interference fringes on-site $m = 6$ when $t_1/t_2 = 1.2$, whereas there are five of them when reducing the coupling ratio to $t_1/t_2 = 0.8$. More generally, it shows that $N_A = m$ for $t_1/t_2 = 1.2$ and $N_A = m - 1$ for $t_1/t_2 = 0.8$. In contrast, we do not observe such a change on sublattice $B$, where there are always $N_B = m$ constructive-interference fringes per site, regardless of the topological phase (Fig. 2c, d).

To explain this striking feature in the LDOS maps near the edge, we focus on a semi-infinite SSH chain and model the edge as an infinite potential barrier. Backscattering of the delocalised waves on the edge then leads to the LDOS[34]

$$\rho_A(m,k) \propto 1 + \cos(2km + \delta_A(k) + \pi) \quad (2)$$

$$\rho_B(m,k) \propto 1 + \cos(2km + \pi), \quad (3)$$

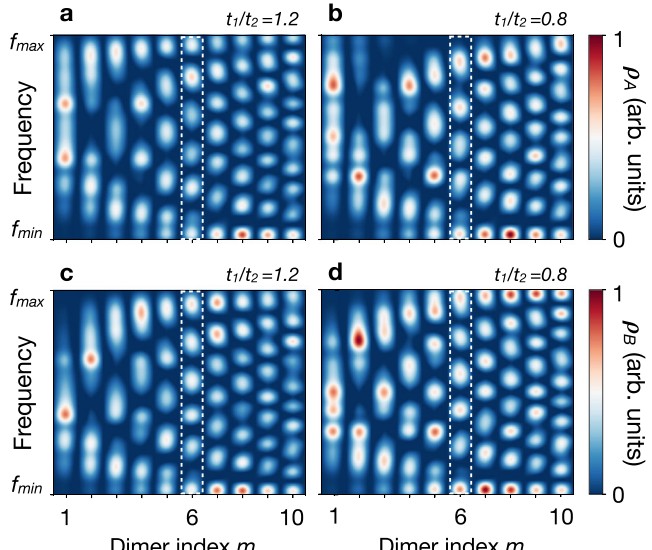

**Fig. 2 Experimental LDOS maps.** Resolution of the lower band between frequencies $f_{min} = 7.23$ GHz and $f_{max} = 7.45$ GHz (see Fig. 1c). **a** LDOS $\rho_A$ of sublattice A for $t_1/t_2 = 1.2$. **b** LDOS $\rho_A$ of sublattice A for $t_1/t_2 = 0.8$. The number $N_A$ of LDOS interference fringes changes identically on each dimer from $N_A = m$ in **a** to $N_A = m - 1$ in **b** (see e.g. the in white dashed box at dimer index $m = 6$). **c** LDOS $\rho_B$ of sublattice B for $t_1/t_2 = 1.2$. **d** LDOS $\rho_B$ of sublattice B for $t_1/t_2 = 0.8$. The number $N_B$ of LDOS interference fringes remains identical in **c** and **d** (see e.g. in the white dashed box at dimer index $m = 6$).

which reproduces very well the experimental LDOS maps in Fig. 2 (see Supplementary Note 3). The oscillating terms in the right-hand sides describe the LDOS fluctuations induced by the edge. The wavelength of the oscillations on both sublattices relates to the backscattering wavevector $2k$. Such $2k$-wavevector oscillations are often referred to as (frequency-resolved) Friedel oscillations, with reference to the charge density oscillations that screen charged impurities in metals[35]. It varies with the frequency at which we probe the cavity through the dispersion relation $f_-(k)$. The wavelength of the oscillations is then a spectral measurement and does not imply the topology of the frequency band. For instance, the oscillations in $\rho_B$ only depend on the backscattering wave-vector and give rise to similar interference patterns in the two topological regimes, as observed experimentally in Fig. 2c, d. In contrast, the oscillations in $\rho_A$ imply the additional phase shift $\delta_A(k) = 2\text{Arg}[h(k)]$ (see ref. [34]).

The phase shift $\delta_A$ further leads to dramatic modifications in the LDOS interference patterns. In particular, the number of constructive-interference fringes $N_A$ is given by the variation of the phase $\varphi_A = 2km + \delta_A + \pi$ in Eq. (2) over the lower frequency band. It reads

$$N_A = \frac{1}{2\pi}\int_{f_{min}}^{f_{max}} \frac{\partial\varphi_A}{\partial f_-}df_- = \int_0^\pi \frac{dk}{2\pi}\left(2m + \frac{\partial\delta_A}{\partial k}\right) = m + w. \quad (4)$$

This sum rule shows that the scattering phase shift $\delta_A$ relates an observable quantity of the delocalised waves, $N_A$, to their topological winding $w$. In particular, the number of interference fringes depends on the site index $m$, but the topological invariant shifts the interference fringes identically on all sites. Remarkably, if the winding number $w$ depends on the choice of unit cell[36,37], this arbitrary choice is however included in the labelling of the dimers $m$, such that their sum yields the observable quantity $N_A$. The sum rule then explains the uniform change of $N_A$ observed in the LDOS maps in Fig. 2a, b for the winding numbers $w_> = 0$ and $w_< = -1$. Therefore, the LDOS maps reveal direct evidence of the

band topology of the delocalised waves in the 1D microwave insulator.

The phase shifts of wave functions also play a central role in scattering physics, because they relate to the number of (virtual) bound states at a potential barrier. Fundamental theorems, such as Levinson theorem or Friedel sum rule, show that, for given wave functions, the number of (virtual) bound states change with the depth of the barrier[35,38,39]. Similarly here, the bulk-boundary correspondence can be rephrased as a relation between the scattering phase shift $\delta_A$ and the number $\mathcal{N}_A$ of bound states localised at the potential barrier of the edge. Since $\delta_A(\pi) - \delta_A(0) = 2\pi w$ (see Eq. (4)), we readily find

$$\delta_A(0) - \delta_A(\pi) = 2\pi\mathcal{N}_A. \qquad (5)$$

We stress that the number of bound states here changes with the topological transition, whereas the strength of the potential barrier remains the same, in sharp contrast with usual defects bound states. This change results from an intrinsic property of the delocalised waves and the potential barrier at the edge only acts as a natural interferometer that reveals their topology through the scattering phase shift. If the phase shift variation has been measured through $N_A$ in the LDOS maps of sublattice A (Fig. 2a, b), we have also resolved the $\mathcal{N}_A$ midgap bound states localised at the edge (inset of Fig. 1d). Thus, both sides of Eq. (5) are observable independently, and our measurements also bring evidence of the bulk-boundary correspondence. This demonstrates an efficient method to test this key concept of gapped topological systems through the LDOS in the experiments.

**Wavefront dislocations in the LDOS**. Now we show that the change of $N_A$ observed in the LDOS maps arises as a ubiquitous wave phenomenon and is the signature of a wavefront dislocation in the LDOS. Topological defects in waves rely on generic assumptions that do not involve the wave equation, and so they are ubiquitously involved in branches of physics as various as electromagnetism, optics, acoustics, fluid physics, astrophysics, and condensed matter physics[2–8,10,11,40,41]. The wavefront dislocations are associated with the topological phase singularities of wavefields in a space of at least dimension 2.

Here, the microwave photonic insulator is 1D and its topological transition relies on a spectral band crossing in the 1D BZ (Fig. 3a). Nevertheless, the topological transition is driven by the coupling ratio $t_1/t_2$. Thus, it is fully characterised in a 2D space associated with the parameter $\mathbf{s} = (t_1/t_2, k)$. In this parameter space, the spectral bands are $f_\pm(\mathbf{s}) = \pm|h(\mathbf{s})|$ and the eigenstates can be chosen as $|u_\pm(\mathbf{s})\rangle \propto |A\rangle \pm e^{i\theta(\mathbf{s})}|B\rangle$, where $\theta(\mathbf{s}) = \mathrm{Arg}[h(\mathbf{s})]$. The zeroes of $h(\mathbf{s})$ are points where i) the spectral band gap closes and ii) the eigenstate phase $\theta(\mathbf{s})$ becomes ill-defined. This phase singularity in 2D is nothing but a vortex that constrains the surrounding phase texture to wind. The vortex winding is then quantified by a topological index $W_s$, such that $\oint_C \nabla_s\theta \cdot \mathbf{ds} = 2\pi W_s$ along a closed circuit $C$ enclosing the phase singularity. In the SSH model, $\mathbf{s_0} = (1, \pi)$ is the only point where $h(\mathbf{s})$ vanishes (Fig. 3b). This leads to the singularity charge $W_s = 1$ for the counter-clockwise circuit $C$ in Fig. 3c.

The phase singularity in the 2D parameter space is the source of a wavefront dislocation of strength $2W_s$ in the LDOS. We can evidence the dislocation by following the evolution of the LDOS interference patterns through the topological transition. Figure 4a shows the predicted LDOS evolution on a given site of sublattice A ($m = 2$). It exhibits an edge dislocation with two constructive-interference fringes emerging from the core in $\mathbf{s_0}$. Wavefront dislocations are known to occur as the phase singularity of a complex scalar field whose real (or imaginary) part represents a physical quantity[1]. Here, the physical quantity is the LDOS

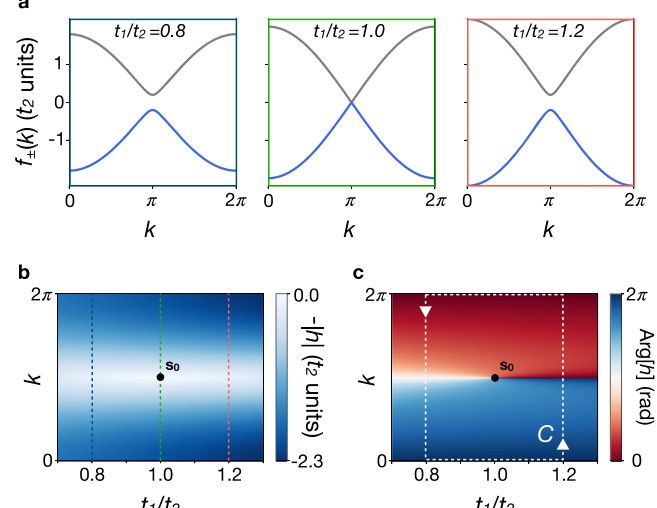

**Fig. 3 Band degeneracy and phase singularity. a** Spectral bands $f_\pm(k) = \pm|h(k)|$ in the 1D BZ for various values of the parameter $t_1/t_2$. The band crossing for $t_1/t_2 = 1$ marks the topological transition between the two insulating regimes. **b** Energy band $f_-(\mathbf{s}) = -|h(\mathbf{s})|$ represented in the 2D parameter space. The spectral degeneracy occurs at point $\mathbf{s_0} = (1, \pi)$. The coloured vertical dashed lines correspond to the 1D spectral bands in **a**. **c** Eigenstate phase $\theta(\mathbf{s}) = \mathrm{Arg}[h(\mathbf{s})]$ in the 2D parameter space. It is singular at the spectral degeneracy point $\mathbf{s_0}$. The winding of the phase along the counter-clockwise circuit $C$ leads to the singularity charge $W_s = 1$.

fluctuations defined as $\Delta\rho_A = -\mathrm{Im}\Delta G_A/\pi$ (oscillating term in Eq. (2)), so that the complex scalar field is the scattering Green function $\Delta G_A$ that describes the delocalised waves backscattering on the edge of the microwave insulator

$$\Delta G_A(m, \mathbf{s}) \propto -ie^{i\varphi_A(\mathbf{s})}, \qquad (6)$$

where $\varphi_A(\mathbf{s}) = 2km + \delta_A(\mathbf{s}) + \pi$ (see Supplementary Note 3). The scattering phase shift $\delta_A(\mathbf{s}) = 2\mathrm{Arg}[h(\mathbf{s})]$ maps the phase singularity of the eigenstates into the phase of the scattering Green function. The latter effectively describes a plane wave ($e^{i2km}$) passing through a vortex ($e^{i\delta_A(\mathbf{s})}$) in the 2D parameter space. This effective vortex perturbs the surrounding phase of the wave in such a way that, for the counter-clockwise Burgers circuit $C$ in Fig. 4a, the phase accumulated by the scattering Green function satisfies

$$2\pi W_d \equiv \oint_C \nabla_s\varphi_A(\mathbf{s}) \cdot \mathbf{ds} = 4\pi W_s \qquad (7)$$

The phase variation is $2\pi$-quantised because $\Delta G_A$ must be single valued to describe observable LDOS fluctuations along the circuit $C$. Thus, the number of additional interference fringes required to fulfil the phase variation along the Burgers circuit $C$ is $W_d$. In analogy with Burgers' vectors whose length provides the dislocation strength for atomic planes in solids, $W_d$ is the strength of the wavefront dislocation. Since $W_s = 1$ in the SSH model, there are $W_d = 2$ additional interference fringes emerging from the dislocation core, as shown in Fig. 4a. It is worth stressing that although the expression of the phase $\varphi_A$ depends on the choice of the unit cell, its variation over $C$ does not and is observable.

To confirm this prediction, we measure the LDOS on the site $m = 2$ of sublattice A for 20 values of the coupling ratio $t_1/t_2$ between 0.2 and 2.0. Since the LDOS is resolved as a function of the frequency $f_-(k)$ instead of the wave vector $k$ in our experiments, we do not expect $W_d$ but $W_d/2$ interference fringes emerging from the dislocation core (Fig. 4b). Figure 4c

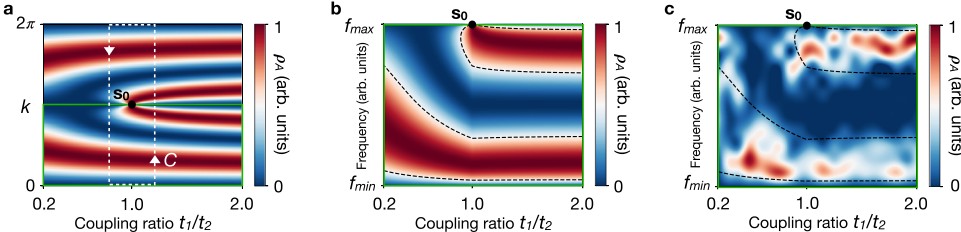

**Fig. 4 Wavefront dislocation in the LDOS. a** Theoretical LDOS $\rho_A$ in the 2D parameter space on site $m = 2$. The phase singularity in $\mathbf{s_0} = (1, \pi)$ yields a wavefront dislocation of strength $W_d = 2$ in the counter-clockwise Burgers circuit $C$. **b** Theoretical LDOS $\rho_A$ on site $m = 2$ in the 2D space $(t_1/t_2, f_-(k))$, where $f_-(0) = f_{min}$ and $f_-(\pi) = f_{max}$. Only one interference fringe emerges from the dislocation core in this frequency representation of the LDOS. **c** Experimental LDOS $\rho_A$ resolved on site $m = 2$ for the lower frequency band. The black dashed lines mark the theoretical wavefronts expected for $\Delta\rho_A = 0.5$ expected from **b**. It shows that one constructive-interference fringe emerges in the LDOS nearby $\mathbf{s_0}$.

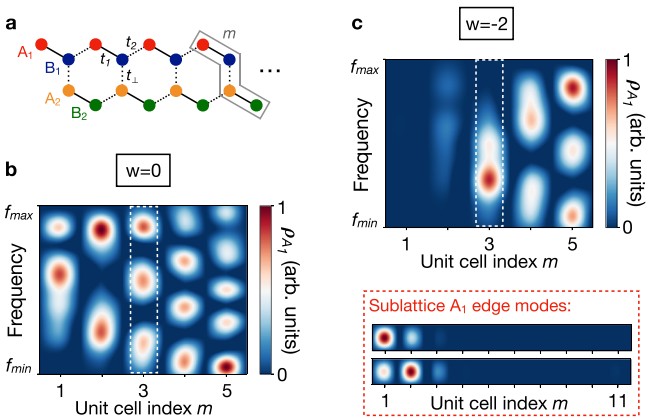

**Fig. 5 Experimental LDOS of a four-band 1D microwave insulator. a** Illustration of a 1D insulator made of two coupled SSH chains and allowing larger winding numbers. The unit cell $m$ involves four sublattices $A_1$, $B_1$, $A_2$, and $B_2$. They are connected through three coupling strengths $t_1$, $t_2$ and $t_\perp$. **b** LDOS $\rho_{A_1}$ of the delocalised waves resolved on leftmost edge of sublattice $A_1$ for $t_1 = 0.8\,t_\perp$, $t_2 = 0.2\,t_\perp$ and $t_\perp = 117$ MHz where $w = 0$. We have resolved the third band, located above the central gap and centred at 7.5 GHz. The microwave crystal made of 11 unit cells of 4 resonators. There are $m$ interference fringes on each unit cell. The number of fringes confirms the sum rule validity for $m \geq 2$ (see Supplementary Note 4). **c** Same as panel b for $t_1 = 0.2\,t_\perp$ and $t_2 = 0.8\,t_\perp$ where $w = -2$. The number of interference fringes has homogeneously shifted by $-w = 2$ units (see e.g. unit cell $m = 3$ in the white dashed box), thus revealing a wavefront dislocation of strength $-w = 2$ between the two topological regimes. The red dashed box shows the LDOS $\rho_{A_1}$ of the $-w = 2$ midgap modes polarised on sublattice $A_1$. The LDOS is in arbitrary units—red (blue) colour means 1 (0). It confirms that the two midgap modes are localised at the leftmost edge on sublattice $A_1$, thus demonstrating the bulk-edge correspondence.

experimentally confirms that the number $N_A$ of constructive-interference fringes changes from one to two at the dislocation core. This is also in agreement with the LDOS change on site $m = 2$ shown in Fig. 2a, b. This observation reveals the wavefront dislocation at the topological transition that causes the uniform change in the number of fringes $N_A$ in the LDOS interference pattern.

The number of interference fringes in the LDOS can also reveal the bulk topology of 1D insulators with winding numbers larger than $|w| = 1$. To demonstrate this beyond the SSH insulator, we consider the (quasi-)1D microwave lattice depicted in Fig. 5a. By varying the coupling ratio $t_1/t_2$, one can experience two distinct topological regimes associated with the winding numbers $w = 0$ and $w = -2$ (see Supplementary Note 4). Remarkably, we show theoretically that the scattering phase shift in the LDOS also leads to the sum rule (4) on sublattice $A_1$ for $m \geq 2$, where

$N_{A_1} = m + w$ (see Supplementary Note 4). It is further confirmed experimentally by our LDOS measurements reported in Figs. 5b,c, where we observe that $N_{A_1}$ shifts by two units between $w = 0$ and $w = -2$. This change in the wavefronts of the interference field is direct evidence of a dislocation of charge $W_d = 4$ associated with the topological transition (see Supplementary Note 4). Moreover, the simultaneous resolution of the corresponding number of midgap edge modes on sublattice $A_1$ allows us to demonstrate experimentally the bulk-edge correspondence in that microwave insulator (inset in Fig. 5d).

## Discussion

We have probed the band topology of 1D photonic insulators through the standing-wave interference pattern in the LDOS resulting from backscattering on a boundary. We have shown that the uniform change in the number of interference fringes at the topological transition is a measurement of the dislocation strength and then of the eigenstate phase singularity. This 2D phase singularity constrains the 1D winding numbers of the two nonequivalent insulators as $W_s = w_> - w_<$ (see Fig. 3). Although there is a gauge choice in the definition of the 1D winding numbers, their difference is gauge invariant and the uniform change in the number of interference fringes characterises unambiguously the change of band topology at the transition. Thus, the wavefront dislocation in the LDOS is an observable phenomenon that reveals the topological transition in 1D insulators. We also emphasise that this direct characterisation of the band-structure topology in real space relies here on the monotonic feature of the dispersion relation—see e.g. Eq. (4). For non-monotonic dispersion relations, there exist several scattering wave-vectors at same energy. However, these are usually well resolved from the LDOS in Fourier space, where the topological scattering phase can be resolved too[34,40]. Such a Fourier analysis is what enabled recent experiments to extract real-space wavefront dislocations as manifestation of topological semimetals with non-monotonic dispersion relations[10,11].

The band topology of 1D insulators is also known to affect the electron response to external force fields through phenomena such as the electric polarisation and Bloch oscillations[42–44]. Nevertheless, these phenomena are observable in very specific systems. Bloch oscillations, for instance, are hardly observable with electrons in solids, where impurities are usually detrimental to phase coherence, and so they lead to band topology measurements in cold atoms[19] or coupled electronic circuits[45]. The concept of mean chiral displacement has been also used in photonic or cold atoms experiments to extract topological invariants from the bulk[46–48]. Our approach lies on an universal observable, the LDOS, which is routinely resolved in various kinds of systems[26–33]. Thus, we expect that topological defects in real-space LDOS interference can reveal the band topology in

experiments involving propagating waves of very different natures.

In addition to the band topology through wavefront dislocations, the LDOS also leads to the resolution of midgap modes localised at boundaries. This enabled us to test the bulk-boundary correspondence through a single observable and, thus, a single experiment. This efficient approach could then shed light into breakdowns of the bulk-boundary correspondence, as recently reported in systems where the number of bound states may no longer be provided by the bulk topological invariant[22,49].

## Methods

Experimental realisation of the quasi-1D photonic insulators: Each dielectric microwave resonator (see Fig. 1a, b) is made of ZrSnTiO ceramics (radius $r = 3$ mm, height $h = 5$ mm, with an index of refraction $n_r \approx 6$) and supports a fundamental transverse-electric mode $TE_1$ of bare frequency $\nu_0 = 7.435$ GHz. This mode spreads out evanescently, so that the coupling strength can be controlled by adjusting the separation distance between the resonators[23]. As shown in Fig. 1b, the lattice consists of two coupled sublattices A and B with staggered coupling strengths $t_1$ and $t_2$. The corresponding resonator separations are denoted $d_1$ and $d_2$. In our experiments, the coupling strengths $t_{1,2}$ can be typically adjusted from 10 to 115 MHz which corresponds to separations $d_{1,2}$ of 16 mm and 7 mm, respectively.

## Data availability

The data that support the findings of this study are available from the corresponding author upon reasonable request.

## Code availability

The codes that support the findings of this study are available from the corresponding author upon reasonable request.

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

## Acknowledgements
M.B. and F.M. acknowledge fruitful discussions with Ulrich Kuhl. C.D. acknowledges the support of Idex Bordeaux (Maesim Risky project 2019 of the LAPHIA Programme).

## Author contributions
M.B. and F.M. performed the experiments, while P.L.D. and C.D. provided theoretical support.

## Competing interests
The authors declare no competing interests.
