## [Peer Review File · Nature Communications]

REVIEWER COMMENTS

Reviewer #1 (Remarks to the Author):

The authors propose a method to probe the band topology through the interference of bulk state wavefunctions. They show that the band topology of 1D insulators can be revealed by the defect-induced wavefront dislocations associated with the topological transition. The theoretical proposal has been verified by the nice experimental results of a 1D microwave photonic insulator. I therefore think this work would have strong potential for publication in Nature Communications if the following questions and comments can be addressed properly.

1. The authors show that there are $N_A=m$ and $m-1$ fringes at A sublattice in the trivial and topological phase, while the number of fringes at B sublattice is always $N_B=m$. The LDOS in Fig. 2 shows a good agreement with the theory for $m>1$, however, there are several LDOS maxima at $m=1$. For example, there are 2 LDOS maxima at $m=1$ in Fig. 2a. Can the authors explain the disagreement?
2. In the tight-binding model with nearest neighboring hopping, there is no difference between an edge and a bulk defect. In real systems, the coupling between resonators is beyond the nearest neighbors. Would there be any difference for the wavefront dislocation induced by an edge and a bulk defect?
3. The whole paper studies 1D insulators, while the dimensionality is not specified in the title. Therefore, the wavefront dislocation scheme is supposed to work for higher dimensional insulators. If this is the case, the authors should discuss the generalization of wavefront dislocation to 2D and 3D insulators. If not, the dimensionality should be specified in the title.
4. The derivation of Eq. (6) is shown in the Supplementary Information that is not referred in the main text.

Reviewer #2 (Remarks to the Author):

The authors study topological transition of the one-dimensional Su-Schrieffer-Heeger (SSH) model in a lattice of microwave resonators. The authors show that, in a two-dimensional plane spanned by the momentum and the ratio of couplings strengths of the SSH model, there is a phase singularity (vortex) of the local density of state (LDOS). The main claim of this paper is that the authors find a relation between the wavefront dislocation and the topological phase transition.

Even though I see that there are some interesting findings, I find one problem/difficulty in the paper, and hence cannot recommend the publication of the paper. I explain my point below in detail.

One problem/difficulty I find with the paper is the procedure to go from the momentum space picture to the frequency space picture. In Fig.3c, the singularity in $\text{Arg}[h]$ in the plane of $(t_1/t_2, k)$ has been depicted. The existence of such a singularity in $(t_1/t_2, k)$ is clear from the definition of the winding number and the topological phase transition. The challenging point is that k space is not the space where the authors can resolve. Instead, the LDOS is resolved in the frequency space. If the singularity exists in $(t_1/t_2, \text{frequency})$ space in an observable form is a question to be asked separately. Since k -space picture is rather well understood, I think the step to go from the k -space to frequency space should be the major challenge and novelty in this paper. I feel that this step has not been explained enough. (I rather received the impression that the authors do not want to discuss this step.)

For example, in Eq.(4), the authors change the frequency-space integral to k-space integral. This equation, in my opinion, cannot be understood in a logical way. Two lines above Eq.(4), the authors define n_A as an integer which satisfies the equation $2\pi n_A = 2km + \delta_A + \pi$. For a given value of m , there are several values of k which gives rise to an integer value of n_A . For example, assuming $\delta_A = 0$ (I mean, let's say we are on B sublattice), when $m = 3$, the combinations $(n_A, k) = (1, \pi/6), (2, \pi/2), (3, 5\pi/6), (4, 7\pi/6), (5, 3\pi/2),$ and $(6, 11\pi/6)$ satisfy the given relation. These are discrete points in (n_A, k) plane. Now, in Eq.(4), the authors take a *derivative* of n_A with respect to f_- . How can one take a derivative where n_A is defined only on discrete points in k space? In any case, since n_A is an integer, and since an integer cannot change continuously, any derivative of n_A should be either 0 or undefined. Even after resolving this point there is another difficult step; there is some relation between n_A and k , but relation between n_A and f_- is still unclear at this point. The authors seem to be assuming that f_- is a monotonically increasing (or decreasing) function of k and f_- takes maximum and minimum at $k = 0$ and π . This is true for SSH model, but it is an assumption in general. This assumption also casts a question about the generality of the authors' result; if the result holds only for the specific SSH model considered here or if the result is valid for a more general one dimensional Hamiltonian respecting a chiral symmetry. Are the authors' method also applicable to systems with longer-ranger hopping where winding number can become larger than 1?

Another place where going from momentum space to frequency space can become an issue is the discussion before the concluding paragraph. The first subtlety here, which I am sure the authors are aware of, is that it is not possible to experimentally confirm the relation involving the phase such as Eq.(7), because one cannot reconstruct ϕ_A beyond the range $[0, \pi]$, which is the principal value of the inverse-cosine function. I do understand that what the authors want to count is not the integral Eq.(7) itself, but the difference of the number of the interference fringes in Fig.4a along $t_1/t_2 < 1$ and $t_1/t_2 > 1$, and this is related to the winding number.

Now, to go from Fig.4a to Fig.4b, the authors again need to assume certain properties of the model. The authors do not explain how one moves from the k-space picture to the frequency-space picture. The relation similar to Eq.(7) would perhaps not hold in frequency space picture anymore. What is the assumption that the authors are making to claim that there should be $Wd/2$ fringes in frequency space? For example, Fig.4a is symmetric around $k = \pi$. Perhaps the dispersion is also symmetric around $k = \pi$. Do we need to assume this symmetry to claim something about the number of fringes in frequency space? (Such a symmetry is perhaps related to the fact that the model is in class BDI, instead of just AIII, in the ten-fold way classification.) Or is just a monotonic behavior from $k = 0$ to π enough? The authors say there should be $Wd/2$ fringes in frequency space; again, is this true also for longer-ranged models with winding larger than 1?

Finally, Fig.4c is the experimental data corresponding to the theoretical one in Fig.4b. I perfectly understand that experiments are always not as clean as theoretical calculations, and data can contain various forms of noise, but I should say that Fig.4c does not convince me that there is a dislocation at the point marked s_0 . The point s_0 looks nothing special in Fig.4c; there are many other points which are similar to the point s_0 in Fig.4c. Is there any quantitative way to confirm that there indeed is a wavefront dislocation in the point s_0 in Fig.4c?

My understanding from the title/abstract/introduction is that Fig.4b and Fig.4c are the main results of the paper. Since I find that the derivation (or the validity) of Fig.4b is unclear, and I am not so convinced by Fig.4c, I cannot recommend the publication of the paper in its current form in Nature Communications.

One final (minor) comment. The authors claim that the band topology is "mainly evidenced through"

the existence of edge states. Although it is true that many of the experiments have focused on edge state detection, there are also some notable exceptions. Restricting to one dimensional systems similar to the SSH model, there is a method of measuring the mean-chiral displacement which have been employed in various systems:

<https://www.nature.com/articles/ncomms15516>

<https://www.nature.com/articles/s41534-019-0159-6>

<https://arxiv.org/abs/2002.09528>

In microwave regime, the Zak phase has also been detected in:

<https://journals.aps.org/prb/abstract/10.1103/PhysRevB.97.041106>

Reviewer 1:

The authors propose a method to probe the band topology through the interference of bulk state wavefunctions. They show that the band topology of 1D insulators can be revealed by the defect-induced wavefront dislocations associated with the topological transition. The theoretical proposal has been verified by the nice experimental results of a 1D microwave photonic insulator. I therefore think this work would have strong potential for publication in Nature Communications if the following questions and comments can be addressed properly.

1. The authors show that there are $N_A = m$ and $m - 1$ fringes at A sublattice in the trivial and topological phase, while the number of fringes at B sublattice is always $N_B = m$. The LDOS in Fig. 2 shows a good agreement with the theory for $m > 1$, however, there are several LDOS maxima at $m = 1$. For example, there 2 LDOS maxima at $m = 1$ in Fig. 2a. Can the authors explain the disagreement?

We thank the Referee for this remark. It helped us to realise that using the wording “*LDOS maxima*” could be the source of a misunderstanding. What is central is the phase winding of the interference field, and so the number of interference fringes between two zeroes. The two LDOS maxima on site $m = 1$ belong to the same interference fringe. Therefore, we have rephrased the paragraph before Eq.(4) and now speak of “*interference fringes*” instead of “*LDOS maxima*” throughout the manuscript.

2. In the tight-binding model with nearest neighboring hopping, there is no difference between an edge and a bulk defect. In real systems, the coupling between resonators is beyond the nearest neighbors. Would there be any difference for the wavefront dislocation induced by an edge and a bulk defect?

Our initial motivation was to focus on the edge to relate locally a bulk scattering property — the scattering phase shift and its wavefront dislocation — to the midgap edge states, and so rephrase the bulk-edge correspondence in terms of physical observables. Whether a bulk defect can also reveal the nontrivial topology through a wavefront dislocation is a natural issue, and we are grateful to the Referee for pointing this out.

First, it is interesting to note that the onsite potential we consider may also be regarded as a bulk defect, even within a nearest-neighbour approximation. One can realise it, for instance, when assuming that the potential is very weak, while the edge corresponds to the limit of very large potential. The potential strength further affects the non-zero component t_{BB} of the T -matrix, which is local in space (see Supplementary Information). Thus, it does not change the LDOS fluctuations described by bare Green functions and always leads to the same wavefront dislocation in the interference pattern. This shows that even a bulk defect can also induce the same wavefront dislocation.

What the potential strength does change, however, is the intensity of the LDOS fluctuations. For instance, a very weak bulk potential V_B leads to $t_{BB} \simeq V_B$ at first order in perturbation theory. The LDOS fluctuations $\Delta\rho_A \propto V_B \cos(2km + \delta(k) + \pi)$ can then be very small, thus making the wavefront dislocation difficult to resolve in the experiment.

Second, it is also true that the coupling between resonators can go beyond the nearest-neighbour approximation in real systems. In our microwave experiments, for instance, there is some coupling between second nearest-neighbours (see e.g. Ref. [23]). This can be limited in the pure 1D geometry of the SSH chain but plays a non-negligible role in the new quasi-1D experiments we have performed to experience higher winding numbers (see Supplementary Information and discussions with Reviewer 2). Nevertheless, such a coupling leads to diagonal matrix elements scaling with the identity matrix in the Bloch Hamiltonian $H(k)$. If it can modify the spectrum, it does not change the eigenstates and so does not change the band structure topology. Thus, the wavefront dislocations observed in the experiment show good agreement with our nearest-neighbour theoretical description.

3. The whole paper studies 1D insulators, while the dimensionality is not specified in the title. Therefore, the wavefront dislocation scheme is supposed to work for higher dimensional insulators. If this is the case, the authors should discuss the generalization of wavefront dislocation to 2D and 3D insulators. If not, the dimensionality should be specified in the title.

We thank the Referee for this remark. If one can expect LDOS topological defects in insulators of higher dimensions — the key assumption being the existence of phase singularities in the bulk wave-functions — we do not study it in the manuscript. To avoid any misunderstanding, we have followed the Referee’s suggestion and specified the space dimension in the title.

4. The derivation of Eq. (6) is shown in the Supplementary Information that is not referred in the main text.

We thank the Referee for pointing out this issue. This is now referred in the revised version of the manuscript.

Reviewer 2:

The authors study topological transition of the one-dimensional Su-Schrieffer-Heeger (SSH) model in a lattice of microwave resonators. The authors show that, in a two-dimensional plane spanned by the momentum and the ratio of couplings strengths of the SSH model, there is a phase singularity (vortex) of the local density of state (LDOS). The main claim of this paper is that the authors find a relation between the wavefront dislocation and the topological phase transition.

Even though I see that there are some interesting findings, I find one problem/difficulty in the paper, and hence cannot recommend the publication of the paper. I explain my point below in detail.

One problem/difficulty I find with the paper is the procedure to go from the momentum space picture to the frequency space picture. In Fig.3c, the singularity in $\text{Arg}[h]$ in the plane of $(t_1/t_2, k)$ has been depicted. The existence of such a singularity in $(t_1/t_2, k)$ is clear from the definition of the winding number and the topological phase transition. The challenging point is that k space is not the space where the authors can resolve. Instead, the LDOS is resolved in the frequency space. If the singularity exists in $(t_1/t_2, \text{frequency})$ space in an observable form is a question to be asked separately. Since k -space picture is rather well understood, I think the step to go from the k -space to frequency space should be the major challenge and novelty in this paper. I feel that this step has not been explained enough. (I rather received the impression that the authors do not want to discuss this step.)

We thank the Referee for this remark. We do not believe that there is any challenging procedure to go from momentum space to frequency space. They relate to one another through the dispersion relation $f(k)$. Furthermore, the phenomenon of standing-wave interference in the LDOS has long been known to be a way to reconstruct the dispersion relation in the experiments [see e.g. PRB (R) 57, 6858 (1998); Science 317(5835), 219–222 (2007)].

In our opinion the major challenge and novelty in this paper is the prediction and the observation of an interference pattern as an original manifestation of a topological property in the bulk of an insulator. To do so, we indeed use an explicit model Hamiltonian for the system. The relation between k -space and the frequency domain is nothing but the dispersion relation $f_-(k)$ of this model Hamiltonian, and we don't believe there is something more profound here. Since our theoretical expressions agree very well with the experimental data, it is very reasonable to claim that our observation of the wavefront dislocation in the LDOS is direct evidence of the existence of the phase singularity in $(t_1/t_2, \text{frequency})$ space for the SSH chain.

For example, in Eq.(4), the authors change the frequency-space integral to k -space integral. This equation, in my opinion, cannot be understood in a logical way. Two lines above Eq.(4), the authors define n_A as an integer which satisfies the equation $2\pi n_A = 2km + \delta_A + \pi$. For a given value of m , there are several values of k which gives rise to an integer value of n_A . For example, assuming $\delta_A = 0$ (I mean, let's say we are on B sublattice), when $m = 3$, the combinations $(n_A, k) = (1, \pi/6), (2, \pi/2), (3, 5\pi/6), (4, 7\pi/6), (5, 3\pi/2),$ and $(6, 11\pi/6)$ satisfy the given relation. These are discrete points in (n_A, k) plane. Now, in Eq.(4), the authors take a *derivative* of n_A with respect to f_- . How can one take a derivative where n_A is defined only on discrete points in k space? In any case, since n_A is an integer, and since an integer cannot change continuously, any derivative of n_A should be either 0 or undefined.

We thank the Referee for pointing out this issue. Thanks to the Referee's remark, we have realised that the way n_A was introduced in the manuscript could be a source of confusion: "*Let us focus, for instance, on the LDOS maxima in Eq. (2). They correspond to the specific wavefronts $2n_A\pi = 2km + \delta_A + \pi$, where n_A is an integer*". What we meant is that the LDOS have maxima when n_A is integer, that is, for integer values of n_A . Of course, it varies continuously, can be differentiated, and its variation provides the number of fringes we are interested in. Therefore, there is nothing wrong with dn_A in Eq. (4). To avoid any further misunderstanding with the reader, we have rephrased the paragraph before Eq.(4) in the revised manuscript.

Even after resolving this point there is another difficult step; there is some relation between n_A and k , but relation between n_A and f_- is still unclear at this point. The authors seem to be assuming that f_- is a monotonically increasing (or decreasing) function of k and f_- takes maximum and minimum at $k = 0$ and π . This is true for SSH model, but it is an assumption in general. This assumption also casts a question about the generality of the authors' result; if the result holds only for the specific SSH model considered here or if the result is valid for a more general one dimensional Hamiltonian respecting a chiral symmetry. Are the authors' method also applicable to systems with longer-ranger hopping where winding number can become larger than 1?

We thank the Referee for these remarks. First, we respectfully do not believe that there is anything unclear here. Our experiment is a realisation of the SSH model. As a matter of fact we support our experimental observations with the SSH model. Thus,

our LDOS description in Eqs. (2) and (4) are theoretical expressions derived from the SSH band structure (1). They readily involve the specific symmetries and dispersion relation of the SSH model, for which $f_-(k)$ is monotonic between $k = 0$ and $k = \pi$. This theoretical description aims to support the experimental observations, and it seems that they are in perfect agreement.

Second, we recall that our claim was to demonstrate the existence of topological defects in the LDOS of an insulator as a novel manifestation of its nontrivial bulk topology, which further allows experimental evidence of the bulk-boundary correspondence. To demonstrate this existence we focus on a realisation of the SSH model. Whether wavefront dislocations exist in more insulators with chiral symmetry will not change our results and conclusions, and so it goes beyond the scope of our initial experiment.

Nevertheless, we agree with the Referee that testing the validity of our results beyond the SSH model would also highlight the generality of our method and so strengthen our results. We have followed the Referee's suggestion and performed another experiment on a more general 1D Hamiltonian with chiral symmetry. If our experiment does not allow us to consider distant hopping terms, we managed to realise an insulator that falls into the symmetry class of Hamiltonian (1) and for which the winding number can take values up to $w = 2$. Accordingly, we observe a LDOS wavefront dislocations of strength $W_d/2 = 2$, and we also support these observations theoretically. This shows that our results extend beyond the SSH model where the winding number can become larger than 1. We are very grateful to the Referee for this remark that helped us to improve the strength of the manuscript. This issue is now discussed in the revised manuscript in connection to Fig. 5. It is detailed in the last section "Generalisation to larger winding number" of the SI, where we additionally show that the low-frequency physics of such a system effectively realises a SSH insulator coupling distant neighbours.

Another place where going from momentum space to frequency space can become an issue is the discussion before the concluding paragraph. The first subtlety here, which I am sure the authors are aware of, is that it is not possible to experimentally confirm the relation involving the phase such as Eq.(7), because one cannot reconstruct ϕ_A beyond the range $[0, \pi]$, which is the principal value of the inverse-cosine function. I do understand that what the authors want to count is not the integral Eq.(7) itself, but the difference of the number of the interference fringes in Fig.4a along $t_1/t_2 < 1$ and $t_1/t_2 > 1$, and this is related to the winding number.

We respectfully disagree with the Referee. The integral in Eq. (7) is the charge of a vortex — circulation of the phase gradient — and it is well known to be observable. We do not pretend to resolve the phase φ_A , whose local value is arbitrary, but we measure the phase variation along a closed path, which is gauge invariant (cf. Aharonov-Bohm phase, Berry phase, ...). Since the edges of the Brillouin zone are wavefronts along which, by definition, the phase does not vary, the change in the number of fringes in Figs.2a,b and 4a,b,c between $t_1 < t_2$ and $t_1 > t_2$ is already a direct measure of the dislocation strength. It is a measure of the vortex charge that perturbs the wavefronts, and the vortex charge is nothing but the winding of the phase given in Eq.(7). It is the charge of the wavefront dislocation. The gauge-invariant nature of Eq.(7) is precisely what makes the wavefront dislocations observable (See e.g. Refs. [2,4,8,9,10,11,12] for similar discussions).

Now, to go from Fig.4a to Fig.4b, the authors again need to assume certain properties of the model. The authors do not explain how one moves from the k-space picture to the frequency-space picture. The relation similar to Eq.(7) would perhaps not hold in frequency space picture anymore. What is the assumption that the authors are making to claim that there should be $W_d/2$ fringes in frequency space? For example, Fig.4a is symmetric around $k = \pi$. Perhaps the dispersion is also symmetric around $k = \pi$. Do we need to assume this symmetry to claim something about the number of fringes in frequency space? (Such a symmetry is perhaps related to the fact that the model is in class BDI, instead of just AIII, in the ten-fold way classification.) Or is just a monotonic behavior from $k = 0$ to π enough? The authors say there should be $W_d/2$ fringes in frequency space; again, is this true also for longer-ranged models with winding larger than 1?

As discussed above, the knowledge of the system symmetry, which protects the band-structure topology, and of the dispersion relation, which is encoded into the scattering wave-vector of the energy-resolved LDOS, enable a straightforward connection between Figs.4a and 4b. Furthermore, our additional theoretical and experimental studies for which we find $W_d/2 = 2$ as a manifestation of a winding number larger than 1 show that our results extend beyond the SSH model.

Finally, Fig.4c is the experimental data corresponding to the theoretical one in Fig.4b. I perfectly understand that experiments are always not as clean as theoretical calculations, and data can contain various forms of noise, but I should say that Fig.4c does not convince me that there is a dislocation at the point marked s_0 . The point s_0 looks nothing special in Fig.4c; there are many other points which are similar to the point s_0 in Fig.4c. Is there any quantitative way to confirm that there indeed is a wavefront dislocation in the point s_0 in Fig.4c?

Figure 4c shows an extra interference fringe emerging/ending nearby s_0 . This is by definition a wavefront dislocation, as announced by the title of the paper. The point s_0 is nothing but the theoretical point introduced in Figs.4a and 4b. It specifies where we expect the dislocation core in the experiment. Then Figure 4c just confirms that the experimental dislocation core lies nearby s_0 , as predicted. We have added some wavefronts as guide for the eyes in Fig.4b and 4c to make the comparison easier between theory and experiment.

My understanding from the title/abstract/introduction is that Fig.4b and Fig.4c are the main results of the paper. Since I find that the derivation (or the validity) of Fig.4b is unclear, and I am not so convinced by Fig.4c, I cannot recommend the publication of the paper in its current form in Nature Communications.

Here, we kindly disagree with the Referee. We recall that Fig.2 already presents the main result of the paper. The change in the number of interference fringes in Figs.2a,b already measures the charge of the wavefront dislocation, exactly in the same way as counting the number of atomic planes along a Burgers' circuit provides the charge of a structural dislocation. Then, Figures 4b,c further enable the visualisation of the dislocation core and confirm that it lies at the topological transition. We also stress that our experimental observations are *all* supported by *transparent* theoretical descriptions and they both show very good agreements. We finally emphasise that following the Referee's suggestion, we even extended our method beyond the SSH model for larger winding numbers. Once again our experimental observations agree very well with our theoretical description, which also proves the validity of our previous results for the SSH chain.

One final (minor) comment. The authors claim that the band topology is "mainly evidenced through" the existence of edge states. Although it is true that many of the experiments have focused on edge state detection, there are also notable exceptions. Restricting to one dimensional systems similar to the SSH model, there is a method of measuring the mean-chiral displacement which have been employed in various systems:

<https://www.nature.com/articles/ncomms15516>

<https://www.nature.com/articles/s41534-019-0159-6>

<https://arxiv.org/abs/2002.09528>

In microwave regime, the Zak phase has also been detected in:

<https://journals.aps.org/prb/abstract/10.1103/PhysRevB.97.041106>

We thank the Referee for mentioning these notable exceptions. We have included them in the concluding paragraph, along with the 1990s works of Resta and Vanderbilt on the mean displacement involved in the modern theory of polarisation. Finally, we would like to stress that the theoretical description of the mean displacement in the works mentioned by the Referee also requires unavoidably the band-structure knowledge of the specific system under investigation. For instance, the evaluation of the mean chiral displacement in [<https://arxiv.org/abs/2002.09528>] deeply relies on chiral symmetry and the explicit knowledge of the polaritonic dispersion relation $E(k)$.

REVIEWER COMMENTS

Reviewer #1 (Remarks to the Author):

I carefully read the revised manuscript and response. I think the authors addressed all my comments and questions. For the comments of Reviewer 2, the main concern is the accordance between Fig. 4b (theory) and Fig. 4c (experiment). As far as I am concerned, the gap (deep blue region) between the two fringes (red stripes) in Fig. 4c is clear. The new experiment of the 1D microwave insulator with winding -2 is interesting and provides further support to the work. Before being able to make a final decision, I have a few more comments for the authors:

1. The main conclusion of this work is that there are $N_A=m+w$ fringes in the topological phase. I am confused about what happens when N_A is negative or zero for small m . For the SSH model with $w=-1$, there is one fringe at $m=1$ in Fig. 2b but $N_A=0$ in this case. However, in Fig. 5c for the second model with $w=-2$, there is no fringe for $m=1$ and 2 where $N_A=-1$ and 0, respectively. I know the theory can well reproduce the experimental LDOS maps even for negative and zero N_A in Fig. S3 and S6. But it seems that $N_A=m+w$ only works for positive N_A . I think this is a very important point that needs to be clarified.

2. Why there is no fringe in Fig. S6a but two (or one?) fringes in Fig. S6b?

3. Some typos: The "transverse-electric" is abbreviated as (TE1) in line 60 but is cite as (TE) in the following. In line 199, the LDOS should be reported in Fig. 5b, c but not Fig. 5a, b.

Reviewer #2 (Remarks to the Author):

I thank the authors for the revision, especially including analysis for models with winding number greater than 1. What I feel is still lacking from the paper is the statement about the range of validity of the authors' method. In reply to my comment, the authors write

"Our experiment is a realisation of the SSH model. As a matter of fact we support our experimental observations with the SSH model. Thus, our LDOS description in Eqs. (2) and (4) are theoretical expressions derived from the SSH band structure (1). They readily involve the specific symmetries and dispersion relation of the SSH model, for which $f_-(k)$ is monotonic between $k = 0$ and $k = \pi$. This theoretical description aims to support the experimental observations, and it seems that they are in perfect agreement."

I think it should be made clear if the result the authors derived is a general result which can be applied to a general class of 1D photonic systems with certain symmetries, or if it only applies to the SSH model. From what the authors wrote above, it seems that their result is a model-specific result relying on the fact that the authors have the SSH model. On the other hand, the authors also confirm the validity of their result for models with winding number greater than 1 in the newly added discussion, implying that their result is more general than just the SSH model.

Is the authors' method applicable to any 1D photonic systems with a chiral symmetry? (I doubt it, because, in general, the energy is not a monotonic function of wavenumber in the presence of chiral symmetry.) I want to emphasize that I perfectly understand that the authors' method works for particular models that the authors analyzed in the text, such as the SSH model and the newly added model. However, I strongly believe that the authors should discuss how general (or how restrictive)

the presented method is. (Or do the authors want to say that the range of validity is not important here, and what is important is that the method works for the SSH model and other specific models that the authors confirmed?)

Other comments:

In response to my comment that, from the LDOS one cannot fully determine the phase ϕ_A , the authors responded:

"We respectfully disagree with the Referee. The integral in Eq. (7) is the charge of a vortex - circulation of the phase gradient - and it is well known to be observable. We do not pretend to resolve the phase ϕ_A , whose local value is arbitrary, but we measure the phase variation along a closed path, which is gauge invariant (cf. Aharonov-Bohm phase, Berry phase, ...). Since the edges of the Brillouin zone are wavefronts along which, by definition, the phase does not vary, the change in the number of fringes in Figs.2a,b and 4a,b,c between $t_1 < t_2$ and $t_1 > t_2$ is already a direct measure of the dislocation strength. It is a measure of the vortex charge that perturbs the wavefronts, and the vortex charge is nothing but the winding of the phase given in Eq.(7). It is the charge of the wavefront dislocation. The gauge-invariant nature of Eq.(7) is precisely what makes the wavefront dislocations observable (See e.g. Refs. [2,4,8,9,10,11,12] for similar discussions)."

My claim was that by looking only at LDOS, which depends on the cosine of the phase, one cannot reconstruct the phase because cosine does not distinguish between θ and $-\theta$. Now, the authors claim that the phase gradient or variation is observable. I am not convinced in this context. Phase gradient is observable, for example, through the observation of the velocity field. In the language of Bose-Einstein condensate, the phase of the condensate wavefunction is observable if one has an access to the local velocity of the condensate. Observing LDOS is similar to observing only at the real part of the wavefunction, from which one has this ambiguity of θ and $-\theta$. The authors quote many papers to show that the charge of a vortex, or the wavefront dislocation, is observable. If I look at, for example, Ref[8] by Berry et al., they propose to observe the phase gradient through the observation of "water velocity". I totally agree that if one has an access to observable such as the water velocity, the phase (gradient) is observable. However, my understanding of the current paper is that the authors only measure LDOS, and I do not see how the phase gradient can be observed by only looking at LDOS.

Regarding Fig.4c, the authors now added theoretical dashed lines as a guide to eyes. With these dashed lines, I do see some similarity between Fig.4b and 4c. My personal feeling is still that Fig.4c can be interpreted to reflect a structure like Fig.4b, only if one is very positive. But I understand that comparison of figures such as 4b and 4c is rather subjective and I do not want to comment further against the authors' claim.

Reviewer 1:

I carefully read the revised manuscript and response. I think the authors addressed all my comments and questions. For the comments of Reviewer 2, the main concern is the accordance between Fig. 4b (theory) and Fig. 4c (experiment). As far as I am concerned, the gap (deep blue region) between the two fringes (red stripes) in Fig. 4c is clear. The new experiment of the 1D microwave insulator with winding -2 is interesting and provides further support to the work. Before being able to make a final decision, I have a few more comments for the authors:

1. The main conclusion of this work is that there are $N_A = m + w$ fringes in the topological phase. I am confused about what happens when N_A is negative or zero for small m . For the SSH model with $w = -1$, there is one fringe at $m = 1$ in Fig. 2b but $N_A = 0$ in this case. However, in Fig. 5c for the second model with $w = -2$, there is no fringe for $m = 1$ and 2 where $N_A = -1$ and 0, respectively. I know the theory can well reproduce the experimental LDOS maps even for negative and zero N_A in Fig. S3 and S6. But it seems that $N_A = m + w$ only works for positive N_A . I think this is a very important point that needs to be clarified.

We thank the reviewer for this remark. We also believe that it deserves some clarification. On the theoretical ground, the sum rule that we derive has a clear domain of validity. It holds when $N_A = m + w \geq 0$.

For the SSH model, the sum rule is then mathematically valid for all sites. For instance, we can verify in Fig. S3c that there is no theoretical issue on site $m = 1$ when $w = -1$. Figure S3c represents the theoretical LDOS map when $w = -1$. It shows one fringe on site $m = 1$. However, this is not in contradiction with the sum rule $N_A = m + w = 0$. Indeed, integer N_A quantifies the winding of the cosine phase of the LDOS oscillations — see Eq. (4). Since the LDOS never reaches the value +1 (dark red colour in Fig. S3c), the cosine phase does not wind and so $N_A = 0$ on $m = 1$. This shows that the sum rule mathematically holds on site $m = 1$, even when $w = -1$. Nevertheless, we agree with the reviewer's comment about Fig. 2b. There are several sources of disturbance in the experiment (noise, losses, ...). Then it is not clear from the experimental LDOS map in Fig. 2b whether the sum rule is verified on site $m = 1$ when $w = -1$, although theory and experiment show a very good agreement otherwise.

As for the second model, we mathematically demonstrate that the sum rule holds if and only if $m \geq 2$ — please see Eqs. (S31) and (S32). Thus, theory and experiment cannot be compared on $m = 1$, where $N_A = -1$ when $w = -2$. However, they show a very good agreement when $m \geq 2$, even on site $m = 2$ where $N_A = 0$ when $w = -2$. We have rephrased some sentences to specify this issue more clearly in the caption of Fig. 5 and in the associated discussion in the main text before the conclusion.

We hope that this clarifies the point raised by the reviewer.

2. Why there is no fringe in Fig. S6a but two (or one?) fringes in Fig. S6b?

We are grateful to the reviewer for pointing this out. As also mentioned in the discussion above, Figure S6a represents the theoretical LDOS fluctuations. It relies on the analytical expression in Eq. (S34), which is valid if and only if $m \geq 2$. Thus, it cannot be compared to the experimental signal on site $m = 1$ in Fig. S6b (now Fig. S6c in the revised version).

To avoid any misunderstanding, we have made new theoretical panels that no longer show the site $m = 1$ in Fig. S6 (this site was previously left blank because no prediction was made, and not because we predicted no signal). We also specify the domain of validity explicitly in the caption of Fig. S6, as well as in the paragraph below Eqs. (S31) and (S32).

3. Some typos: The “transverse-electric” is abbreviated as (TE1) in line 60 but is cite as (TE) in the following. In line 199, the LDOS should be reported in Fig. 5b, c but not Fig. 5a, b.

We are also grateful to the reviewer for mentioning these typos. We have made the changes in the revised manuscript.

Reviewer 2:

I thank the authors for the revision, especially including analysis for models with winding number greater than 1. What I feel is still lacking from the paper is the statement about the range of validity of the authors' method. In reply to my comment, the authors write: "Our experiment is a realisation of the SSH model. As a matter of fact we support our experimental observations with the SSH model. Thus, our LDOS description in Eqs. (2) and (4) are theoretical expressions derived from the SSH band structure (1). They readily involve the specific symmetries and dispersion relation of the SSH model, for which $f_-(k)$ is monotonic between $k = 0$ and $k = \pi$. This theoretical description aims to support the experimental observations, and it seems that they are in perfect agreement."

I think it should be made clear if the result the authors derived is a general result which can be applied to a general class of 1D photonic systems with certain symmetries, or if it only applies to the SSH model. From what the authors wrote above, it seems that their result is a model-specific result relying on the fact that the authors have the SSH model. On the other hand, the authors also confirm the validity of their result for models with winding number greater than 1 in the newly added discussion, implying that their result is more general than just the SSH model.

Is the authors' method applicable to any 1D photonic systems with a chiral symmetry? (I doubt it, because, in general, the energy is not a monotonic function of wavenumber in the presence of chiral symmetry.) I want to emphasize that I perfectly understand that the authors' method works for particular models that the authors analyzed in the text, such as the SSH model and the newly added model. However, I strongly believe that the authors should discuss how general (or how restrictive) the presented method is. (Or do the authors want to say that the range of validity is not important here, and what is important is that the method works for the SSH model and other specific models that the authors confirmed?)

We thank the reviewer for these comments on the validity of our approach. Our latest experiment indeed shows that it is relevant beyond the SSH insulator. As far as we understand, the reviewer expects us to specify to what extent this is general. In particular, the reviewer is concerned with the monotonic feature of the dispersion relation in our experimental systems.

We would like to avoid any misunderstanding here, and it may be worth recalling that if we do not experience non-monotonic dispersion relations, this is not because we dismiss the reviewer's comment. Non-monotonic dispersion relations imply coupling strengths between distant neighbours, which we cannot realise in our experimental platform. This is the only reason why we have had to think about a new class of insulators to experience larger winding numbers within a nearest neighbour description. Then it turns out that the class of 1D chiral insulators that we can experience — N coupled SSH chains — leads to monotonic dispersion relations between $k = 0$ and $k = \pi$. This is what allows the variable change in Eq. (4) and so what sets the validity of our experimental results.

This being said, we would like to make two more comments.

First, we do believe that one can still resolve the band-structure topology associated with non-monotonic dispersion relations. Then the only subtlety is the existence of several scattering wave-vectors at given energy. However, it is known that the scattering wave-vectors can be well resolved in Fourier space from the LDOS — see e.g. [PRB (R) 57, 6858 (1998)], [Science 317, 219–222 (2007)] — where the topological winding of the scattering phase can be resolved too [10,34,40]. It is then possible to filter specific wave-vectors of interest in Fourier space and extract the corresponding real-space interference pattern. This is precisely what enabled recent experiments to reveal the band-structure topology of 2D semimetals associated with non-monotonic dispersion relations [10, 11]. For the sake of clarity, we now mention it explicitly in the first paragraph of the conclusion.

Second, we also recall that our initial experimental realisation of the SSH insulator did not pretend to establish the generality of a novel approach — experiments are not general in essence. Instead, it aimed at demonstrating a novel unanticipated manifestation of the nontrivial bulk topology of insulators. Whether it is more general is of course an issue of interest. And we could show that our approach extends to a class of chiral insulators beyond the SSH insulator. But this does not change our initial conclusions that, we believe, already have a large impact on their own:

1) Our initial work experimentally demonstrates a relation between the topological index of a band insulator and a new observable: the LDOS. In addition to the conceptual interest, this observable is universal and routinely resolved in various experimental platforms (cold atoms, photonics, electronic materials, ...). In contrast, previously known observables relating to the band-structure topology of insulators were restricted to specific types of experimental systems. The phenomenon of Bloch oscillations, which is known to reveal the band-structure topology of 1D insulators, cannot be observed in electronic materials due to short coherence times. Neither can it be emulated in our microwave platform. Thus our novel approach based on a

universal observable importantly allows one to study the band-structure topology in these experimental platforms too. We also emphasise that this new relation to the LDOS further allows us to investigate the bulk state topology and the edge states, thus providing the experimental demonstration of the bulk-edge correspondence. To the best of our knowledge, this was not possible with previously known observables.

2) Our initial work experimentally generalises a fundamental concept of quantum scattering theory. The scattering phase shifts were known to depend on the scattering potential. We show that they additionally depends on the topological phase singularities of the scattering wave functions. This is important because the impurity problem is heavily involved in condensed matter physics (Friedel, RKKY, Anderson localisation, Kondo, DMFT, ...).

3) It is commonly thought that probing of the band-structure topology requires the study of the states *dynamics* (e.g. Bloch oscillations in 1D) in response to some external forces (e.g. time-dependent gauge field) — see e.g. RMP 2010 by Niu et al. In contrast, our initial work experimentally proves that this is not necessary. Indeed, the LDOS fluctuations consist of a *stationary* interference phenomenon, which is conceptually different. For instance, it involves a density observable instead of a displacement- or current-based observable.

4) Our initial experimental work shows that it is possible to observe the phase singularities/vortex of wave functions through a wavefront dislocation in the density. As acknowledge by the reviewer below, this is rather counter-intuitive with respect to our experience of vortices in condensed matter phases associated with broken U(1) gauge symmetry. Wavefront dislocations have also long been thought unobservable in quantum mechanics. For instance, Berry and collaborators initially thought that *"In quantum mechanics these phase singularities are unobservable [...] not only the intensity but all observable quantities are unaffected"* [8]. Thus, not only our work proves that wavefront dislocations are observable in the density associated with quantum mechanical wave functions, but it further bridges the field of topological insulators to the broad audience physics of singular waves (fluids, sound, electromagnetism, oceanic tides, singular optics, ...).

Other comments:

In response to my comment that, from the LDOS one cannot fully determine the phase ϕ_A , the authors responded:

"We respectfully disagree with the Referee. The integral in Eq. (7) is the charge of a vortex - circulation of the phase gradient - and it is well known to be observable. We do not pretend to resolve the phase A, whose local value is arbitrary, but we measure the phase variation along a closed path, which is gauge invariant (cf. Aharonov-Bohm phase, Berry phase, ...). Since the edges of the Brillouin zone are wavefronts along which, by definition, the phase does not vary, the change in the number of fringes in Figs.2a,b and 4a,b,c between $t_1 < t_2$ and $t_1 > t_2$ is already a direct measure of the dislocation strength. It is a measure of the vortex charge that perturbs the wavefronts, and the vortex charge is nothing but the winding of the phase given in Eq.(7). It is the charge of the wavefront dislocation. The gauge-invariant nature of Eq.(7) is precisely what makes the wavefront dislocations observable (See e.g. Refs. [2,4,8,9,10,11,12] for similar discussions)."

My claim was that by looking only at LDOS, which depends on the cosine of the phase, one cannot reconstruct the phase because cosine does not distinguish between θ and $-\theta$. Now, the authors claim that the phase gradient or variation is observable. I am not convinced in this context. Phase gradient is observable, for example, through the observation of the velocity field. In the language of Bose-Einstein condensate, the phase of the condensate wavefunction is observable if one has an access to the local velocity of the condensate. Observing LDOS is similar to observing only at the real part of the wavefunction, from which one has this ambiguity of θ and $-\theta$. The authors quote many papers to show that the charge of a vortex, or the wavefront dislocation, is observable. If I look at, for example, Ref[8] by Berry et al., they propose to observe the phase gradient through the observation of "water velocity". I totally agree that if one has an access to observable such as the water velocity, the phase (gradient) is observable. However, my understanding of the current paper is that the authors only measure LDOS, and I do not see how the phase gradient can be observed by only looking at LDOS.

We thank the reviewer for this comment. However, we do not claim that the phase variation is observable. Equation (7) states that the phase variation *along a closed path* is observable. The notion of cycle is crucial here. This is precisely what makes the Aharonov-Bohm or Berry phase gauge-invariant. In the manuscript we also recall this is the main condition introduced by Nye and Berry for the observable nature of wavefront dislocations: *Wavefront dislocations are known to occur as the phase singularity of a complex scalar field whose real (or imaginary) part represents a physical quantity [1]*. In other words, the phase singularity of a complex wave function $\psi = \rho e^{i\theta}$ is observable through its real or imaginary part. The key point is that θ has a singularity in a 2D parameter space. We also emphasise that similar wavefront dislocations have been observed

recently in the LDOS [10, 11], which further proves the observable nature of the wave-function phase singularities in the density.

REVIEWERS' COMMENTS

Reviewer #1 (Remarks to the Author):

As far as I am concerned, the authors have carefully addressed the questions and comments of both referees. Provided some questionnaire parts clarified in the revised manuscript, I would like to recommend the manuscript for publication in Nature Communications.